# Similarities and Differences between Clear Cell Tubulo-Papillary and Conventional Clear Cell Renal Cell Carcinoma: A Comparative Phenotypical and Mutational Analysis

**DOI:** 10.3390/diagnostics10020123

**Published:** 2020-02-23

**Authors:** Francesca Giunchi, Tania Franceschini, Elisa Gruppioni, Annalisa Altimari, Elisa Capizzi, Francesco Massari, Riccardo Schiavina, Matteo Brunelli, Guido Martignoni, Michelangelo Fiorentino

**Affiliations:** 1Department of Pathology, University of Bologna, 40138 Bologna, Italy; frachikka@virgilio.it (F.G.); tania.franceschini@studio.unibo.it (T.F.); elisa.gruppioni@aosp.bo.it (E.G.); annalisa.altimari@aosp.bo.it (A.A.); elisa.capizzi@libero.it (E.C.); 2Department of Oncology, University of Bologna, 40138 Bologna, Italy; francesco.massari@aosp.bo.it; 3Department of Urology, University of Bologna, 40138 Bologna, Italy; riccardo.schiavina@unibo.it; 4Department of Pathology, University of Verona, 37129 Verona, Italy; matteo.brunelli@univr.it (M.B.); guido.martignoni@univr.it (G.M.)

**Keywords:** clear cell tubulo-papillary carcinoma, clear cell renal cell carcinoma, next generation sequencing, immunohistochemistry

## Abstract

Background: Clear cell tubulo-papillary renal cell carcinoma (cctpRCC) is characterized by clear cell morphology, but differs from conventional clear cell carcinoma (ccRCC) for its indolent clinical behavior and genetic background. The differential diagnosis between the two is based on histology and immunohistochemistry (IHC). Methods: We performed a comparative case-control histological, IHC, and genetic analysis by next generation sequencing (NGS), to point out the differences in 10 cases of cctpRCC, and six controls of ccRCC with low stage and grade. Results: All 16 cases showed the IHC profile with cytokeratin 7, racemase, and carbonic anhydrase IX expected for the histological features of each tumor type. By contrast, the NGS mutation analysis that covered 207 amplicons of 50 oncogenes or tumor suppressor genes provided conflicting results. Among the 10 cctpRCC cases, eight (80%) were wild type for all of the genes in the panel, while two (20%) harbored *VHL* mutations typical of ccRCC. Three of the six (50%) ccRCC control cases showed expected *VHL* mutations; two (33%) harbored pathogenic mutations in the *p53* or the *CKIT* genes; and one (16%) was wild type. Conclusion: We can assume that histology and ICH are not sufficient for a definitive diagnosis of cctpRCC or ccRCC. Although with a panel covering 50 genes, we found that 80% of cctpRCC were genetically silent; thus, suggesting an indolent biology of these tumors. The differential diagnosis between ccptRCC and ccRCC for the choice of the best therapeutic strategy likely requires the comprehensive evaluation of histology, IHC, and at least *VHL* mutations.

## 1. Introduction

Clear cell tubulo-papillary renal cell carcinoma (cctpRCC) is a rare entity (1–4% of all kidney tumors) that has been recognized by the World Health Organization (WHO) classification since 2004 [1]. cctpRCC displays morphological and immunohistochemical (IHC) features in between conventional clear cell renal cell carcinoma (ccRCC) and papillary renal cell carcinoma (pRCC) [2,3]. The diagnostic clues of cctpRCC include clear cytoplasm with variable tubular/acinar, papillary, and cystic architecture; a linear nuclear array with no atypia; absence of mitoses or necrosis; immunoreactivity for cytokeratin 7 (CK7) and carbonic anhydrase IX (CAIX) (cup-like distribution); and negative immunoreaction for racemase (AMACR) (Figure 1). The combination of the morphological features and the IHC profile is generally considered diagnostic of cctpRCC, in routine practice [1,4]. cctpRCC and conventional ccRCC also harbor completely different molecular alterations. In fact, cctpRCC lacks all the genetic abnormalities typical of ccRCC and pRCC, such as *VHL* alterations (chromosome 3p deletions), or polisomy in chromosomes 7 and 17 [5,6].

From a clinical standpoint, cctpRCC follows an indolent clinical behavior with no local recurrences, and no lymph node or distant metastases reported in the literature, so far. Due to its indolent clinical behavior, some authors have suggested to encounter cctpRCC as a benign lesion or to re-designate it as “clear cell papillary neoplasm of low malignant potential” [7,8,9]. However, in order to change the classification of cctpRCC from a malignant to a low-malignant or benign category, it is necessary to compare histological features, IHC, and mutational profiles of both cctpRCC and ccRCC.

Here we describe a comparative histological, IHC, and next generation sequencing (NGS) mutation analysis of 10 cctpRCC and six small ccRCC.

## 2. Materials and Methods

We selected 10 cctpRCC and 6 control cases of small ccRCC with comparable stage and grade (stage pT1A according to AJCC 2016, nucleolar grade G1, according to WHO 2016), diagnosed in our Institution from 2015 to 2018. The study has been approved by the Institutional Review Board of the Area Vasta Emilia-Romagna Centro with the code PRIORI. Standard operating procedures for tissue fixation, processing, and archival storage were fixation in 4% neutral buffered formalin for 8 to 48 h at room temperature after specimen sectioning; processing in graded alcohols followed by clearing with toluene and 3 serial paraffin baths at 56 °C. Serial 3µm sections were cut for IHC and a single 10-µm section for DNA extraction.

### 2.1. Immunohistochemistry

Immunohistochemistry was performed for CK7, AMACR, and CAIX with the following antibodies and dilutions: CK7, (rabbit monoclonal, clone SP52, prediluted; Ventana Medical Systems, USA), AMACR p504s (rabbit monoclonal, clone SP116, pre-diluted; Ventana Medical Systems ) and CAIX, (rabbit monoclonal, clone TH22, Novocastra diluted 1:100). Reactions were carried out and revealed using an automated Benchmark Ultra instrument (Ventana Roche, Ventana Medical Systems, Inc., Tucson, AZ, USA).

### 2.2. NGS Analysis

Tumor areas of interest with at least 70% tumor cell enrichment were circled, and 10 µm-thick serial sections of the same paraffin block cut in sterility for DNA extraction. Sections were manually microdissected, deparaffinized in xylene, and the DNA isolated using the GeneRead DNA FFPE Kit (Qiagen, Hilden, Germany). DNA was quantified with the Quantifiler^®^ Human DNA Quantification Kit (Thermo Fisher Scientific, Waltham, MA, USA) and 10 ng of DNA were used for library preparation. Next generation sequencing (NGS) analysis was run on an Ion PGMTM System using the Ion AmpliSeqTMCancer Hotspot Panel v2 (ThermoFisher Scientific, Waltham, MA), covering 207 amplicons of the following 50 oncogenes or tumor suppressor genes: ABL1, EGFR, GNAS, KRAS, PTPN11, AKT1, ERBB2, GNAQ, MET, RB1, ALK, ERBB4, HNF1A, MLH1, RET, APC, EZH2, HRAS, MPL, SMAD4, ATM, FBXW7, IDH1, NOTCH1, SMARCB1, BRAF, FGFR1, JAK2, NPM1, SMO, CDH1, FGFR2, JAK3, NRAS, SRC, CDKN2A, FGFR3, IDH2, PDGFRA, STK11, CSF1R, FLT3, KDR, PIK3CA, TP53, CTNNB1, GNA11, KIT, PTEN, VHL. In particular, the panel includes the 124 most common variants of the *VHL* genes including the small deletions. The MET, IDH1, SMARCB1, CDH1, IDH2, PDGFRA, VHL genes have been implicated in RCC.

Successful sequencing of a sample required at least 500, 000 reads with a quality score ≥ Q20. A minimum coverage of 500X with at least 10% frequency was used as cut-off for a variant to be considered true. Sequence alignment and base calling were performed using the Torrent Suite software v.4.4.3 (Thermo Fisher Scientific) taking Human Genome Build 19 (hg19) as the reference. Variant calling was carried out with the Variant Caller v.4.4.3.3 plug-in, using default “Somatic—Low Stringency” settings. Variants were further filtered using Ion Reporter software v.4.4 (Thermo Fisher Scientific).

## 3. Results

All the cases diagnosed as cctpRCC showed small tubules and papillae lined by cells with clear cytoplasm, and small nuclei arranged linearly at the basal membrane. The 10 cctpRCC cases showed a consistent IHC profile (CK7+/AMACR-/CAIX+). The NGS profile showed that 8/10 cases (80%) were wild type for all the investigated genes, while 2/10 (20%) showed VHL gene pathogenic mutations. In particular, one showed a mutation at the splicing site in exon 3 (c.464-1G>A) and the other in exon 2 (c.443_444delTT (p.Phe148fs) of the *VHL* gene (see Figure 2).

Among the six ccRCC control cases, 3/6 cases (50%) showed expected VHL mutations and a coherent IHC profile (CK7−/AMACR−;CAIX+); one (s16%) had consistent IHC profile, but it was wild type for all the genes in the panel, and two (33%) displayed consistent IHC profile, and harbored mutations either in p53 (splicing site exon 5, IVS5+3>A) or the cKIT (splicing site exon 10, c.1594G>A (p.Val532Ile) genes (Figure 2).

Figure 2 shows that the overall concordance between histology and the expected IHC profile was 100% (16/16 concordant cases). The concordance between the expected IHC profile and the expected NGS mutation status was 69% (11/16 cases).

## 4. Discussion

cctpRCC and ccRCC of the kidney share a clear cell morphology but display diverse IHC phenotype, genetic background and clinical behavior. cctpRCC is still recognized as malignant as ccRCC, despite its very mild nuclear atypia, the absence of necrosis and vascular invasion, and the low stage [10,11,12,13,14,15,16,17,18]. The histological differential diagnosis between the two tumor types is based on morphological and IHC features. On molecular grounds, *VHL* gene alterations are common in most ccRCC while they are absent in cctpRCC [18]. In this comparative case-control study, we aimed to assess the phenotypical and molecular differences in cctpRCC and ccRCC with comparable size and cytological atypia.

The 10 cctpRCC in our series showed typical morphological features and IHC (CK+/AMACR−/CAIX+) but two cases harbored *VHL* gene alterations. Therefore, despite the histology and the IHC profile, these two tumors must be considered conventional ccRCC from the molecular standpoint. This assumption is based on the data derived from The Cancer Genome Atlas where the large majority of conventional ccRCC harbored *VHL* alterations [19]. The other eight cctpRCC were wild type for all of the 50 oncogenes or tumor-suppressor genes in our NGS panel. Although our panel is limited to 50 genes, we can be sure that these eight tumors were wild type for *VHL*. With the limitations of our panel, we can assume that these eight cctpRCC tumors were biologically indolent.

Similarly, the six ccRCC case controls showed morphology and an IHC profile consistent with this diagnosis, but only in three cases the NGS analysis confirmed the presence of a *VHL* alteration. Among the other three cases, one was wild type and the other two displayed pathogenic mutations in *p53* and *CKIT* that have been described in ccRCC, also by us [20]. Therefore, the genetic analysis confirmed the diagnosis of ccRCC in all except one case. The wild type status of the remaining tumor does not exclude a large deletion of the chromosome 3p locus, encompassing *VHL*, and could not be detected by sequencing. In addition, clear cell tumors with VHL wild type and mutations in TCEB1 have been recently described to increase HIF stabilization via the same mechanism as VHL inactivation. Unfortunately, TCEB1 was not included in our panel [21].

Our data lead to discuss the role of IHC for the differential diagnosis of renal cell tumors with clear cell morphology. In fact, only 69% of the cases and controls showed morphological, IHC, and genetic concordance. Our results are affected by the limited number of cases compared to the large series that brought to the validation of the IHC diagnostic profiles in the past [22]. However, it should be noted that these IHC results were not cross-validated by wide spectrum mutation analyses.

Although, in a limited number of cases, we can assume that morphology and IHC are not sufficient for a definitive diagnosis of cctpRCC or ccRCC without the confirmation with the genetic profile.

It might be argued that cctpRCC and low-stage/low-grade ccRCC may behave similarly and could be treated both conservatively. Actually, we believe that an accurate differential diagnosis between these two tumors is clinically relevant. If other series with larger NGS panels will confirm that cctpRCCs are de facto genetically silent, these tumors might be re-classified as benign and possibly treated in accordance. In the meantime, we may suggest to include a genetic evaluation, of at least *VHL* in addition to IHC, for the choice of the best therapeutic strategy of these tumors.

## Figures and Tables

**Figure 1 diagnostics-10-00123-f001:**
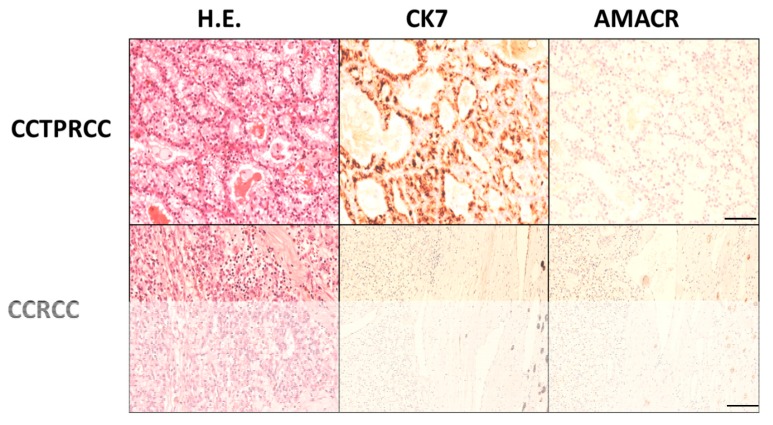
Differential H&E and immunohistochemical features between clear cell tubulo-papillary renal cell carcinoma (cctpRCC) and clear cell carcinoma (ccRCC). cctpRCC shows a peculiar tubular structure with little atypia and nuclei lined over the basal membrane, diffuse cytokeratin 7 (Ck7) reactivity, and negativity for racemase. ccRCC shows irregular nests of clear cells negative for both CK7 and racemase (scale bar 100 microns).

**Figure 2 diagnostics-10-00123-f002:**
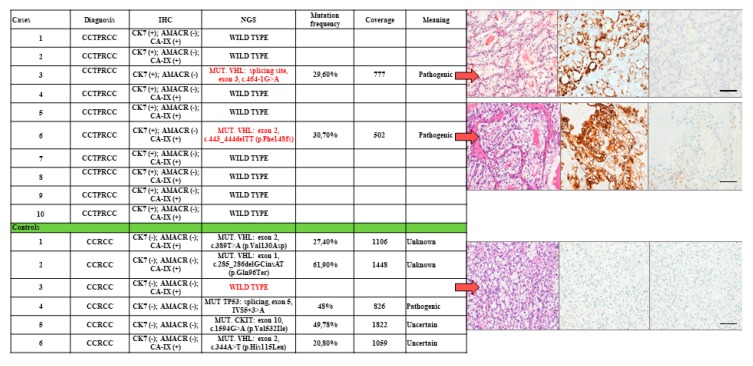
Comparative histological, immunohistochemical, and mutational profiles of 10 cctpRCC cases and six ccRCC controls. The histological and immunohistochemistry (IHC) features of the three cases most discordant with next generation sequencing (NGS) are indicated by arrows (scale bar 50 microns).

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
