# Peer review of "Similarities and Differences between Clear Cell Tubulo-Papillary and Conventional Clear Cell Renal Cell Carcinoma: A Comparative Phenotypical and Mutational Analysis"

_diagnostics, 2020, doi:10.3390/diagnostics10020123_

Round 1
Reviewer 1 Report
In this manuscript, Giunchi et al reported differential diagnostic results from an IHC/NGS combined comparative analysis of cctpRCC versus ccRCC cases, implying insufficiencies of histology and IHC as sole tools for a definitive diagnosis of cctpRCC or ccRCC. This is a clinically insightful study and points out future directions for improved diagnosis of RCC by comprehensive examination of histology, IHC and certain genetic mutations such as VHL. Overall, the manuscript is well written with data presented neatly. Here are the concerns for the authors to clarify and consider to incorporate into revision.
Major points:
1) The relatively small sample size impedes making an affirmative conclusion, so it would be necessary to town down relevant text as appropriate.
2) How reliable and accurate was the NGS used the in the comparative study. The NGS-based diagnosis is somehow inconclusive. For instance, whether or not there is a VHL deletion for the wild-type ccRCC case could not be excluded (page 4 line 136). If this is the case, the biomarker(s) downstream of VHL, such as HIF1a, could be alternatively examined by IHC.
Minor points:
1) What was the selection criteria of including those 50 oncogenes or tumor suppressor genes in the NGS analysis? How many of those genes have been reported to be involved in RCC especially cctpRCC and ccRCC subtypes? It would be informative to briefly mention those relevant to RCC in the Materials/Methods section.
2) Scale bars should be labeled on IHC images.
Author Response
In this manuscript, Giunchi et al reported differential diagnostic results from an IHC/NGS combined comparative analysis of cctpRCC versus ccRCC cases, implying insufficiencies of histology and IHC as sole tools for a definitive diagnosis of cctpRCC or ccRCC. This is a clinically insightful study and points out future directions for improved diagnosis of RCC by comprehensive examination of histology, IHC and certain genetic mutations such as VHL. Overall, the manuscript is well written with data presented neatly. Here are the concerns for the authors to clarify and consider to incorporate into revision.
We thank the reviewer for the appreciation of our work.
Major points:
1) The relatively small sample size impedes making an affirmative conclusion, so it would be necessary to town down relevant text as appropriate.
We have mitigated the conclusions in the last sentences of the Abstract and the Discussion
2) How reliable and accurate was the NGS used the in the comparative study. The NGS-based diagnosis is somehow inconclusive. For instance, whether or not there is a VHL deletion for the wild-type ccRCC case could not be excluded (page 4 line 136). If this is the case, the biomarker(s) downstream of VHL, such as HIF1a, could be alternatively examined by IHC.
The reviewer is right. The NGS panel cannot be completely conclusive but it covers the 124 most common mutation variants of the VHL gene including all the frequent small deletions. This has been now specified in the methods. Only large deletions could be missed by the panel. We did not investigate further the VHL pathway since the overexpression of HIF1a by immunohistochemistry could be difficult to quantitate given its basal expression in non-VHL deleted tumors. In addition, clear cell tumors with VHL wild-type and mutations in TCEB1 have been recently described to increase HIF stabilisation via the same mechanism as VHL inactivation. TCEB1 is not included in our panel. A sentence has been added to the Discussion section as well as a reference.
Minor points:
1) What was the selection criteria of including those 50 oncogenes or tumor suppressor genes in the NGS analysis? How many of those genes have been reported to be involved in RCC especially cctpRCC and ccRCC subtypes? It would be informative to briefly mention those relevant to RCC in the Materials/Methods section.
The Ion AmpliSeqTMCancer Hotspot Panel v2 is a commercially available NGS panel including 50 oncogenes or tumor suppressor genes mutated in most human solid tumors. Seven of them plus VHL have been involved in renal cell cancer. This has been now specified in the Methods.
2) Scale bars should be labeled on IHC images.
Scale bars have been added.
Reviewer 2 Report
The study was aimed to investigate the potential differences and similarities between clear cel tubulo-papillary and conventional clear cell renal cell carcinoma. I have the following suggestions –
- Please use the standard symbols for the unit. e.g., lines 56 and 57 - µm (not um).
- In many places, the abbreviations came first without the full forms or vice versa. e.g., line 35 (IHC), line 39 CK7 and CAIX, line 69 (IHC) and CK7, etc.
- Please correct the spelling for atypia in line 38.
- In figure 2 legends, please make sure this is 200x magnification. I think it’s much smaller than that.
Author Response
The study was aimed to investigate the potential differences and similarities between clear cel tubulo-papillary and conventional clear cell renal cell carcinoma. I have the following suggestions –
- Please use the standard symbols for the unit. e.g., lines 56 and 57 - µm (not um).
We fixed this
- In many places, the abbreviations came first without the full forms or vice versa. e.g., line 35 (IHC), line 39 CK7 and CAIX, line 69 (IHC) and CK7, etc.
We fixed this
- Please correct the spelling for atypia in line 38.
We fixed this
- In figure 2 legends, please make sure this is 200x magnification. I think it’s much smaller than that.
We have added scale bars instead of the magnification